# Phytochemical Composition and Pharmacological Efficacy Evaluation of *Calamintha nepeta*, *Calamintha sylvatica*, *Lavandula austroapennina* and *Mentha piperita* Essential Oils for the Control of Honeybee (*Apis mellifera*) Varroosis

**DOI:** 10.3390/ani14010069

**Published:** 2023-12-24

**Authors:** Roberto Bava, Fabio Castagna, Carmine Lupia, Stefano Ruga, Filomena Conforti, Mariangela Marrelli, Maria Pia Argentieri, Vincenzo Musella, Domenico Britti, Giancarlo Statti, Ernesto Palma

**Affiliations:** 1Department of Health Sciences, University of Catanzaro Magna Græcia, 88100 Catanzaro, CZ, Italy; roberto.bava@unicz.it (R.B.); studiolupiacarmine@libero.it (C.L.); rugast1@gmail.com (S.R.); musella@unicz.it (V.M.); britti@unicz.it (D.B.); palma@unicz.it (E.P.); 2Mediterranean Ethnobotanical Conservatory, 88054 Sersale, CZ, Italy; 3Department of Pharmacy, Health and Nutritional Sciences, University of Calabria, 87036 Rende, CS, Italy; filomena.conforti@unical.it (F.C.); mariangela.marrelli@unical.it (M.M.); g.statti@unical.it (G.S.); 4Department of Pharmacy-Drug Sciences, University of Bari Aldo Moro, 70125 Bari, BA, Italy; mariapia.argentieri@uniba.it; 5Department of Health Sciences, Institute of Research for Food Safety & Health (IRC-FISH), University of Catanzaro Magna Græcia, 88100 Catanzaro, CZ, Italy; 6Nutramed S.c.a.r.l., Complesso Ninì Barbieri, Roccelletta di Borgia, 88021 Catanzaro, CZ, Italy

**Keywords:** honeybee (*Apis mellifera*), *Varroa destructor*, essential oil, contact residual bioassay, *Calamintha sylvatica*, *Calamintha nepeta*, *Lavandula austroapennina*, *Mentha piperita*

## Abstract

**Simple Summary:**

Essential oils (EOs) have been shown to possess several pharmacological properties, among which we can mention antibacterial, antiviral and acaricidal effects. The latter activity is particularly interesting in beekeeping for the control of *Varroa destructor* parasitosis. This research aimed to verify the acaricidal potential of four botanical species belonging to the Lamiaceae family. Specifically, the species tested were *Calamintha sylvatica*, *Calamintha nepeta*, *Lavandula austroapennina* and *Mentha piperita*. The evaluation was conducted by means of residual contact toxicity tests by diluting the EOs in Acetone to achieve concentrations of 2, 1 and 0.5 mg/mL. At the highest concentration, the tests returned a neutralization percentage of 52% for *Calamintha nepeta*, 60% for *Calamintha sylvatica*, 80% for *Lavandula austroapennina* and 68% for *Mentha piperita*.

**Abstract:**

*Varroa destructor* is currently considered the parasite that causes the greatest damage and economic losses to honeybee farms. Its presence is often associated with that of viral and bacterial pathogens, which ultimately leads to colony collapse. Careful control of the parasitic load is therefore necessary to avoid the onset of these events. Although chemical treatments are often in easily and quickly administered formulations, in recent years, there have been increasingly frequent reports of the onset of drug resistance phenomena, which must lead to reconsidering their use. Furthermore, chemical compounds can easily accumulate in the food matrices of the hive, with possible risks for the final consumer. In such a condition, it is imperative to find alternative treatment solutions. Essential oils (EOs) prove to be promising candidates due to their good efficacy and good environmental biodegradability. In this study, the acaricidal efficacy of the EOs of *Calamintha sylvatica* Bromf., *Calamintha nepeta* Savi, *Lavandula austroapennina* N.G. Passal. Tundis & Upson and *Mentha piperita* L., extracted from botanical species belonging to the Lamiaceae family, was evaluated. The test chosen for the evaluation was residual toxicity by contact. The examined EOs were diluted in Acetone to a concentration of 2, 1 and 0.5 mg/mL. At the highest concentration, the EOs demonstrated an acaricidal activity equal to 52% for *C. nepeta*, 60% for *C. sylvatica*, 80% for *L. austroapennina* and 68% for *M. piperita*. Of the EOs tested, therefore, Lavender proves to be a good candidate for subsequent evaluations in semi-field and field studies.

## 1. Introduction

*Varroa destructor* is responsible for varroosis, a serious parasitosis affecting adult and immature honeybees. The life cycles of the *V. destructor* and honeybee are tightly correlated. Seeking out developing honeybees, *V. destructor* parasitizes nurse bees to reach the brood. Adult female *V. destructor* reproduces in capped brood cells by feeding on the fat body of honeybee pupae. Once the developing bee completes its cycle, the *Varroa* mite and her progeny emerge. These mites will then carry on a new biological cycle [1]. Parasitosis has significant negative consequences. The host loses 3% of its body water for every female mite present [2]. The weight loss of nascent *Varroa*-infested honeybees varies from 6.3% to 25%, depending on the number of mites present in the cell during development [2,3]. Honeybees born after parasitization in immature stages emerge with lower levels of protein concentration in the head and abdomen, in the order of 20%, and with a lower concentration of carbohydrates in the abdomen [2]. Honeybees have a shorter life expectancy under these circumstances [4,5]. In addition, *V. destructor* infestation can weaken the honeybees’ immune system, making them more vulnerable to numerous diseases [6,7]. When they emerge, parasitized honeybees have half as much glycogen in their flight muscles as unparasitized bees and start foraging at a young age [2,8]. The so-called parasitic mite syndrome can develop if *V. destructor* is present in a high population. This syndrome is associated with the transmission of various viruses by mites [9,10]. Virus activity is manifested by the appearance of honeybees characterized by a reduced size or atrophy of the abdomen, malformations of wings and legs and a reduction in and dysfunction of various glands [6,11,12]. These changes produce a steady decline in the hive population and a slow colony collapse, often also caused by the introduction of opportunistic secondary diseases [13]. For many years, chemical pest control has proven to be a solution of quick and easy application to control this pest. The development of resistance to many active ingredients and their persistence in honeybee products are the two unsolved problems associated with the use of chemical compounds [14,15]. It has already been extensively established that the phenomenon of resistance exists in Italy, the country where Fluvalinate initially lost its efficacy [16]. More recent reports have been published [17,18]. It is difficult to prevent the onset of drug resistance phenomena; by following the recommended dosages and strictly alternating the various pharmacological products, these phenomena could be limited. With regard to residues, Wallner, 1999, confirmed that coumaphos is moved inside the hive by the honeybee with wax, and this also causes a gradual contamination of the wax sheets that are introduced during the periods when treatment is not carried out [19]. 

Furthermore, there is not much of a barrier between the honeycomb of the supers and the brood nest, where the acaricide treatment is administered. Honeybees, in their work, tend to evenly distribute acaricide residues. Due to these issues, the scientific community is focusing more on alternative control strategies. Consequently, the use of natural compounds such as organic acids is increasing [20]. However, several studies indicate that formic and oxalic acid can be detrimental to honeybee health. For example, the damage and removal of open and capped brood have often been observed [21]. In addition, long-term damage to bees’ digestive and excretory glands and organs [22,23], damage to the queen or even her premature death have been reported [24]. Furthermore, with current treatment protocols, the efficacy of formic acid in controlling varroa mites varies significantly [25,26]. The results of even similar treatments are extremely inconsistent [27]. It is known that treatment efficacy is influenced by several factors. Treatment efficacy depends on the distance between the formic acid volatilization site and the beecomb, as well as whether the acid is added to the hive above or below the brood chamber [28]. The number of brood in the hive [29], the season [30] and the outside temperature [26] have all been proposed as variables that may affect the effectiveness of a treatment.

In light of these considerations, the use of essential oils (EOs) seems to be a more valid remedy, which is generating much interest in the scientific community [31,32,33,34,35]. EOs are natural compounds of a plant origin consisting of mixtures of volatile substances [36,37]. Being composed of several molecules that have different cellular targets, treated pest populations are unlikely to develop drug resistance [38]. Moreover, being natural compounds, they are easily degradable in the environment and do not accumulate in the food matrices of the hive [39,40,41]. EOs from the Lamiaceae family have proven to be particularly effective in controlling *V. destructor* [33,42,43,44]. As proof of this, we can find several pharmaceutical preparations based on thymol, a monoterpene phenol that is highly represented in plants of the genus Thymus [45]. In addition we can cite the example of oregano EO, which, in several efficacy tests conducted by different research groups, has demonstrated a particular acaricidal action [33,42,46]. Through chemical characterization of species of the Lamiaceae family, numerous chemical components, mostly mono- and diterpenoids with a variety of actions against various arthropods, have been identified [47,48]. It is reasonable to speculate that an untapped reservoir of substances with potential antiarthropod action may exist, given the wide distribution of Lamiaceae species in the Mediterranean regions [49]. According to this evidence, it was decided to investigate the possible acaricidal action of four EOs extracted from botanical species native to the Calabria region (southern Italy) and belonging to the Lamiaceae family. In particular, the acaricidal activity of the EOs of *Calamintha nepeta*, *Calamintha sylvatica*, *Lavandula austroapennina* and *Mentha piperita* was tested by means of residual toxicity tests.

## 2. Materials and Methods

### 2.1. Botanical Species Collection

The aerial parts of *C. nepeta*, *C. sylvatica* and *M. piperita* were collected during their balsamic period, specifically in June and July, in natural growing areas in Calabria, southern Italy. *L. austroapennina* EO was provided by “Parco della Lavanda” (Morano Calabro, Cosenza, Italy).

Dr. Carmine Lupia from the Department of Health Sciences of the “Magna Græcia” University of Catanzaro conducted the taxonomic identification. For each botanical species collected, a voucher was deposited at the Mediterranean Ethnobotanical Conservatory, Sersale (CZ), Italy. Specifically, the vouchers are in numbers 2, 3 and 14 of the family Labiatae for *C. nepeta*, *C. sylvatica* and *M. piperita,* respectively. 

### 2.2. EOs Extraction and Analysis

The fresh plant materials were cleaned and subjected to a 2 h steam distillation process to extract the EOs (Albrigi Luigi, Verona, Italy). 

For the chemical analysis, a Trace GC–FID ultra Thermo Finnigan gas chromatograph was utilized. Each distilled EO was solubilized in hexane before analysis, and then 1 μL of EO was injected. A DB-5 (J&W Scientific, Folsom, CA, USA) fused silica capillary column (30 m × 0.25 mm; 0.25 μm film thickness) was employed for the cold on-column injection. These were the chromatographic conditions: 300 °C was the detector temperature; a 4 °C min^−1^ program was used to program the column temperature from 60 °C (5 min isothermal) to 280 °C (15 min isothermal). The carrier gas was hydrogen (2.0 mL/min; 35 kP). The 32-bit computer program Chrom-Card was used to process the data. Based on the total peak regions found in the GC-FID analysis, the composition of the EOs’ components is given as a percentage. No correction factors were used.

The GC-MS analysis was performed using a Hewlett Packard 6890-5973 mass spectrometer (Agilent Technologies, Palo Alto, CA, USA) interfaced with an HP Chemstation. The following were the chromatographic conditions: an injector temperature of 280 °C; a column oven program of 60 °C (5 min isothermal) to 270 °C (30 min isothermal) at 4 °C/min. The carrier gas was helium (1 mL/min flow rate). A capillary column of 30 m × 0.25 mm with a film thickness of 0.25 µm, HP-5 MS, was employed. The MS was operated with the following parameters: vacuum 10-5 torr; ion source temperature, 70 eV; electron current, 34.6 µA. Mass spectra were obtained at 1 scan/s spanning the 40–800 amu range. The electron impact mode was in use at the ion source. The spitless sampling approach was used to inject samples (1 µL). The studied EOs’ chemical composition was ascertained by utilizing reference mass spectra from standard compounds and/or library files, as well as by comparing the GC retention durations of their components with known authentic reference compounds in conjunction with Kovats Indexes (KI) [50].

### 2.3. Toxicological Evaluation against V. destructor

The research took place at the Research Institute for Food Safety and Health (IRC-FSH) of the “Magna Graecia” University of Catanzaro. The acute toxicity of the four candidate natural acaricides was assessed using a residual bioassay, a procedure often used to detect drug resistance and toxicity in arthropods. The four EOs tested were: *C. sylvatica*, *C. nepeta*, *L. austroapennina* and *M. piperita*. With a few minor modifications, Gashout and Guzmán-Novoa’s method was employed [51]. The four EOs and Amitraz (Merck, Darmstadt, Germany, 45323) were diluted in Acetone to concentrations of 2 mg/mL, 1 mg/mL and 0.5 mg/mL. Acetone and Amitraz served as negative and positive controls, respectively. In June and July 2023, the trials were conducted. The mites were obtained from selected apiaries in the province of Catanzaro; the hives were infested by *V. destructor* and had not undergone acaricidal treatment in the previous six months.

Each replicate of the experiment began with mite collection (between 100 and 200 mites each time) from the same apiary using a drone trap frame. From their original colonies, the frames were taken out and brought to the lab. Mite harvesting began with the removal of the wax layer covering the honeybee pupa. Once the cap was removed, the cell was inspected for the presence of mites. The mites found were placed in a Petri dish in which fifth instar larvae and honeybee pupae had previously been placed to feed the mites during the collection phases. Subsequently, the internal surfaces of an Eppendorf tube received the treatments. Specifically, each 1.5 mL Eppendorf tube was filled with 50 µL of EO diluted in Acetone at various concentrations, and the solvent (Acetone, Sigma-Aldrich, Steinheim, Germany) was allowed to dry before adding the mites to the tubes. To dry the Acetone, the test tubes were placed open in an oven and rolled several times on their walls for about 15 min. This also allowed the EO to sprinkle onto the walls of the Eppendorf tube. During the operation, the EO was unlikely to have evaporated before the Acetone, due to its high boiling point, above 200 °C [51]. At the end of the procedure, for each technical replicate (each oil, positive and negative control), five adult female mites were inserted into the test tube. After the introduction of the mites, the tubes were closed and placed in an incubator with a relative humidity of 65% and a temperature of 34 °C. This process was carried out ten times (ten technical replicates) for each EO, as well as for the negative and positive controls. By monitoring mite mortality after 1 h of exposure to each treatment, the relative acute toxicity of each EO was calculated. In particular, the mites were moved from the tubes to a Petri dish and studied under a stereo microscope one hour after exposure. When forced with a brush, mites that did not move from their position were considered dead. Inactive mites were defined as those that moved one or more legs. Inactive and dead mites were considered equally neutralized [32].

### 2.4. Evaluation of the Toxicity of the EOs to Honeybees

Following the method suggested by Bava et al., 2021, the toxicity of the tested EOs to honeybees was assessed [32]. Twenty-five *A. mellifera* worker honeybees were randomly selected. These twenty-five bees were divided and assigned to groups of five individuals which formed the experimental replicates for each of the four EOs and the negative control replicate. Toxicity tests, as described below, were performed and replicated three times. The honeybees selected for the test came from a larger group of honeybees obtained by randomly mixing individuals taken from different combs. In this way, a sample of honeybees of various ages was obtained. In proportion to the quantities used for the toxicity experiments on *V. destructor*, 1.6 mL of the different dilutions of the EOs under investigation were placed in four 50 mL Falcon test tubes, which were then repeatedly rolled on the walls to cover them with liquid and evaporate the Acetone. Once the Acetone of the EO dilution had dried, five bees were placed inside four Falcon tubes, one for each essential oil tested; a Falcon with only Acetone was used as a negative control. For the hour following preparation, the Falcon tubes with the honeybees inside were incubated at a temperature of 34 °C and a relative humidity of 65%. The mortality of the test subjects was assessed after one hour of exposure and in the following 24 and 48 h after placing the honeybees in a cage equipped with feeders loaded with 50% sucrose syrup [52]. 

### 2.5. Statistical Analysis

The complete dataset was transferred to the statistical analysis software for further examination. The program GraphPad PRISM (version 9.0., GraphPad Software Inc., La Jolla, CA, USA) was used to analyze the data. The results are presented as means and SEMs. One-way ANOVA and the Bonferroni test for multiple comparisons were specifically used for the statistical analysis. Statistics were deemed significant for *p* values below 0.05. 

Multivariate analysis was performed using the online software MetaboAnalyst version 5.0 (http://www.metaboanalyst.ca, accessed on 4 November 2023). The multivariate approach of principal component analysis (PCA) was chosen in this study. PCA is a widely used multivariate statistical method for identifying the relationships between the original variables and reducing them to independent principal components [53]. PCA was carried out in order to have a clear overview of the distribution of the phytochemical constituents in the investigated EOs. The data matrix consisted of 12 samples (three analyses for each EO) and 80 variables (the percentage abundance of identified secondary metabolites). First, data integrity was checked, and missing values were replaced by LoDs (1/5 of the minimum positive value of each variable). The data were normalized by log transformation and pretreated through Pareto-scaling. The heat map of a two-way hierarchical clustering analysis (HCA) has also been added to provide a clear visualization of the data table. It was performed using the Euclidean distance measure and Ward’s clustering method.

## 3. Results

### 3.1. Phytochemical Profile

The yield of the EO from *M. piperita* was equal to 0.3% *w*/*w*, while the steam distillation of the two *Calamintha* spp. allowed obtaining yields equal to 0.1% *w*/*w*. 

The phytochemical profile of the four EOs was verified with gas chromatography–mass spectrometry (GC-MS). 

Overall, 80 metabolites were recognized (Table 1).

Menthol (39.91–1.33%), 1,8-cineole (34.09–1.11%) and linalool (26.67–0.12%), followed by menthone (21.67% in MP) and pulegone (20.91–0.78%), were detected in the highest percentages, even if they were differently distributed among the samples. Furthermore, piperitone oxide (18.26% in CS sample), terpinen-4-ol (17.44–0.25%) and borneol (15.24% in LA) were detected at percentages above 15%, while eugenol (14.66%) and linalool acetate (14.67–11.25%) were detected at percentages above 10%.

*C. nepeta* (CN) essential oil was mainly characterized by the presence of 1,8-cineole (34.09%), eugenol (14.66%) and linalool acetate (11.25%), followed by sabinene (6.97%) and linalool (6.64%). α and β-pinene were also detected at percentages above 3%.

The main component of *L. austroapennina* (LA) samples was instead linalool (26.67%), followed by terpinen-4-ol (17.44%), borneol (15.24%) and linalool acetate (14.67%). Camphor was also detected at percentages above 5%.

*C. sylvatica* (CS) was mainly characterized by the high abundance of piperitone oxide (37.70%), pulegone (20.91%) and piperitenone oxide (18.26%). Moreover, iso-menthone (7.5%) and limonene (6.58%) were also detected in good percentages. These two last components were also identified in *M. piperita* EO, whose main constituents, as expected, were menthol (39.91%) and methone (21.67%). 

To obtain a better overview of the data, a Principal Component Analysis (PCA) was performed (Figure 1). PCA reduces the original dataset to a few latent variables or principal components (PCs).

The data matrix consisted of 12 samples (three analyses for each EO) and 80 variables (identified secondary metabolites). Figure 1a reports the score plot obtained by considering the first and the second principal components, which explain 83.6% of the total variance, with PC1 and PC2 accounting for 45.5% and the 38%, respectively.

The score plot highlights that *C. nepeta* (CN) samples are clearly separated from the others by positioning themselves in the top left half of the plot. *C. sylvatica* (CS) and *M. piperita* (MP) samples are located in the lower left half of the plot, while those from *L. austroapennina* (LA) are located in the lower right half of the plot. 

The distribution of EOs’ chemical constituents is clearly visualized in the following heatmap (Figure 2). 

The color scale represents different concentrations of the metabolites, with yellow being the highest and dark blue being the lowest one. The two-way hierarchical clustering analysis (HCA) was performed using the Euclidean distance measure and the Ward’s clustering method.

### 3.2. Acaricidal Activity

Each EO utilized was diluted in Acetone at concentrations of 0.5 mg/mL, 1 mg/mL and 2 mg/mL to assess the treatment’s effectiveness. Also, the active ingredient Amitraz was diluted in Acetone at the same concentrations and used as a positive control. Acetone alone was utilized as the negative control. Ten technical replicates were performed in this experimental design for the various doses. Figure 3 represents the results of the neutralization percentages returned by the EOs tested at the different concentrations.

All EOs tested and Amitraz gave a statistically significant difference compared to the negative control. The tested dilutions of *C. sylvatica* EO are comparable to the lower concentration of the positive control, as are the *C. nepeta* and *M. piperita* EOs. The latter two EOs are also comparable in terms of neutralization to the 1 mg/mL concentration of the positive control. The 0.5 and 1 mg/mL EO concentrations of *L. austroapennina* produced the same control efficacy as Amitraz 0.5 and 1 mg/mL, while the highest concentration of the EO had the same efficacy as the highest concentration of the positive control. All concentrations of *M. piperita* EO gave the same acaricidal efficacy as Amitraz 0.5 and 1 mg/mL.

The following Table 2 summarizes the neutralization percentages of the EOs tested. 

In Figure 4, the tested EOs were compared. In particular, the results of the comparison of the acaricidal activity of the different EOs at different concentrations are shown. *L. austroappenina* at the concentration of 2 mg/mL was statistically more effective than *C. nepeta* at the same concentration. In all other cases, where the concentration was the same, there were no statistically significant differences.

### 3.3. Toxicity towards Honeybee

All EOs at the various concentrations tested showed no toxicity to honeybees. In all experimental replicates, the five individuals placed in each Falcon tube, which had been pre-treated with the EOs under testing, survived the exposure, as well as in the negative control containing only acetone. The honeybees tested, moreover, did not show any abnormal behavior.

## 4. Discussion

The EOs in this study were evaluated using a residual toxicity test, which allowed for an easy and affordable way to confirm the EOs’ acaricidal activity. Numerous published studies have examined the botanical genera under investigation; however, the acaricidal properties of the botanical species tested against *V. destructor* had not been studied, with the exception of *M. piperita*. It should be mentioned that the effectiveness of EO has very seldom been assessed using residual toxicity tests. A residual toxicity test for *Mentha piperita* was employed by Hybl et al. (2021), among other published research [54]. According to their research, the average mortality rate following two hours of exposure was close to 65%, and following four hours, it was 100% [54]. The acaricidal activity was higher than that demonstrated in this study (68% at the highest concentration). The differences in exposure times (1 h in the current study compared to 2 and 4 h considered by Hybl et al., 2021 [54]) and concentrations (2 mg/1 mL acetone in the present study compared to 0.375 µL EO/500 µL acetone used in the study by Hybl et al., 2021 [54]) could be some of the reasons for the discrepancies in the results between the compared studies [54], as well as the variability in the composition of the EOs tested. Using a residual toxicity test, Gashout and Guzmán-Novoa (2009) confirmed the acaricidal activity of different *Mentha* and *Lavandula* species [51]. The authors of this study found that the mortality rate for *V. destructor* was around 20% for both *L. angustifolia* and *M. spicata* when they tested mortality four hours after the initial exposure and used a dose of 1.5 mg/1 mL acetone [51]. In our study, the EOs of *M. piperita* and *L. austroapennina* had an average acaricidal efficacy of 68% and 80%, respectively, at the highest dose examined (2 mg/1 mL acetone). The EOs we examined in this study exhibited varying levels of acaricidal activity, which can be attributable to the extraction process and chemical makeup in addition to the previously indicated exposure time and concentrations. Furthermore, the plant species and organs utilized, the area of origin and the harvest season (balsamic period) all influence these variables [55,56].

The EO of *M. piperita* was characterized by very high concentrations of menthol and methone. The main specific component of *C. nepeta* was 1,8-cineole, whereas the most prevalent components of *C. sylvatica* EO were piperitone oxide, pulegone and piperitenone oxide. Some compounds, such as iso-menthone and limonene, were only detected in the EOs of *M. piperita* and *C. sylvatica*. Finally, the EO of *L. austroapennina* was characterized by the highest abundance of linalool, terpinen-4-ol, borneol and linalool acetate. Among the monoterpenes, linalool and bornyl acetate are known for their high acaricidal activity [57]. The higher presence in the EO of *L. austroapennina* could explain its higher acaricidal activity than that of the other EOs tested. Unquestionably, one reason for the discrepancy in the results across the published research is the variable chemical composition of EOs. To ensure uniform and comparable findings, it would be crucial to standardize the experimental protocols, extraction methods and physicochemical properties of the EOs used in bioassays. At the doses and exposure times at which the mites were tested, all EOs had good effectiveness as compared to the corresponding negative control, and even at 0.5 mg/mL, their efficacy was statistically significant. The most effective EO at the lowest concentration (0.5 mg/mL) was that of *L. austroapennina*. Of all the EOs studied, this latter one showed the most promise and can be employed for additional testing in semi-field and field investigations. However, the efficacy of lavender essential oil, the best of those tested, is not sufficient for the commercialization of a product.

According to the “Guideline on veterinary medicinal products controlling *V. destructor* parasitosis in bees”, the level of control after treatment should preferably be 90% or higher for non-synthetic substances [58]. By developing a formulation that enables more effective medicinal preparation delivery in the hive, this issue might be resolved.

## 5. Conclusions

Several approaches have been suggested to control the *V. destructor* parasite. Despite these efforts, varroosis continues to be a major threat to modern beekeeping. Nowadays, it is necessary to increase the weapons available to beekeepers and limit the phenomena of drug resistance and chemical product residues in the environment and food matrices. EOs can be considered a viable alternative to synthetic drugs for the control of *V. destructor* populations. Among the EOs tested in this study, lavender proved to have the best acaricidal activity, achieving a neutralization rate of 80% at the highest concentration tested. Lavender could therefore be used in subsequent semi-field and field studies. The other EOs demonstrated a modest acaricidal capacity. However, their use should not be ruled out a priori; EOs with low activity could in fact be considered useful for the execution of a mild treatment that also favors the necessary rotation of active ingredients necessary to avoid the onset of drug resistance phenomena. An important consideration that emerges from the comparison of our results with those obtained by other authors is the extreme variability in the result obtained. To decrease this discrepancy between studies, it is recommended to use shared procedures. An essential oil should first of all be verified for its effectiveness through contact residual tests; only once efficacy has been validated should fumigation, full exposure, semi-field and field studies be conducted. Only by following this rigorous methodology can we shed light on the immense potential that plant extracts possess. In conclusion, it can be said that these types of studies lead the way for a new concept of veterinary drug treatment more in line with the green revolution that has been gradually pushing ahead in recent years.

## Figures and Tables

**Figure 1 animals-14-00069-f001:**
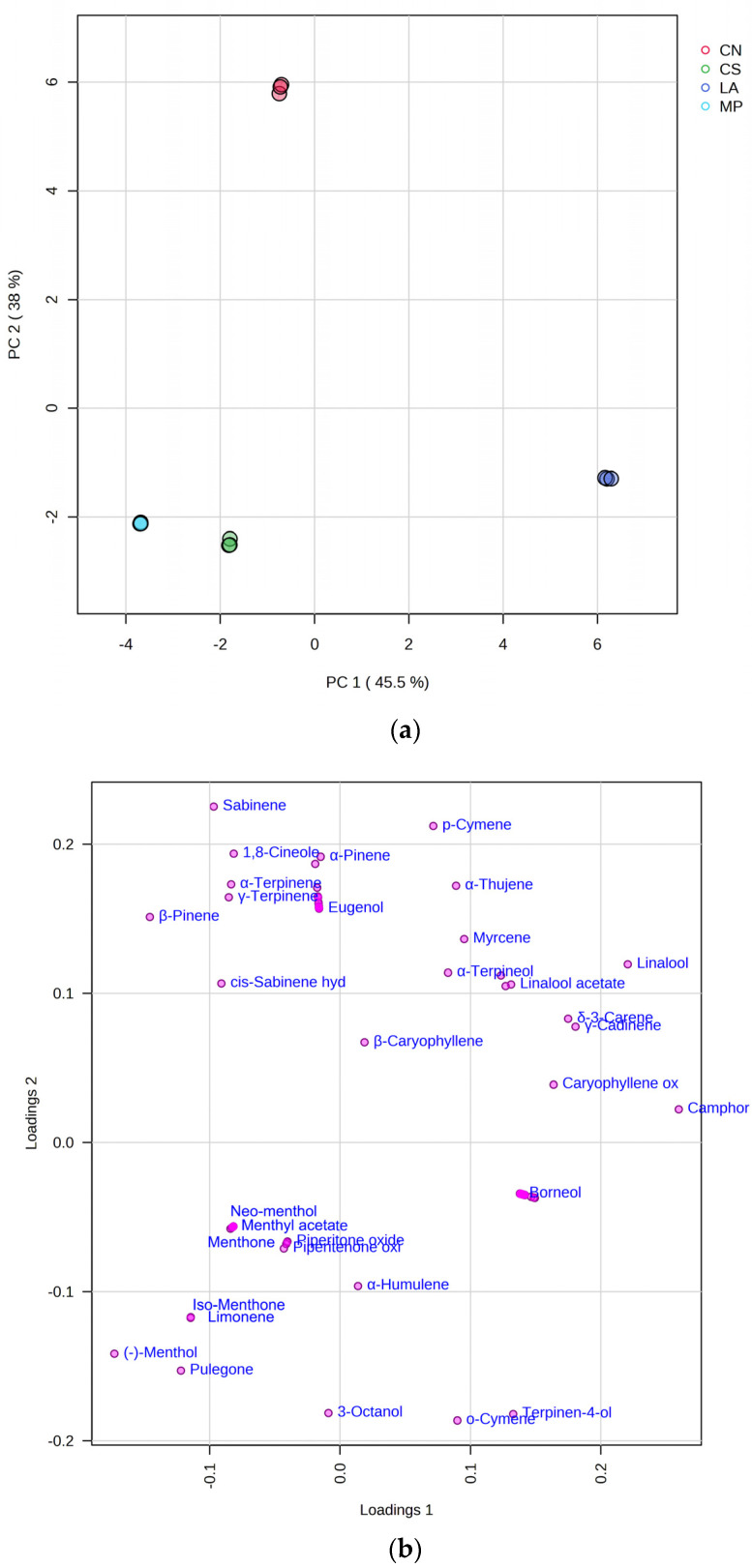
(**a**) PCA scores plot (PC-1 vs. PC-2) based on the chemical composition of the essential oils (EOs). CN: *Calamintha nepeta* Savi; CS: *Calamintha sylvatica* Bromf.; LA: *Lavandula austroapennina* N.G. Passal. Tundis & Upson; MP: *Mentha piperita* L. (**b**) Loadings plot.

**Figure 2 animals-14-00069-f002:**
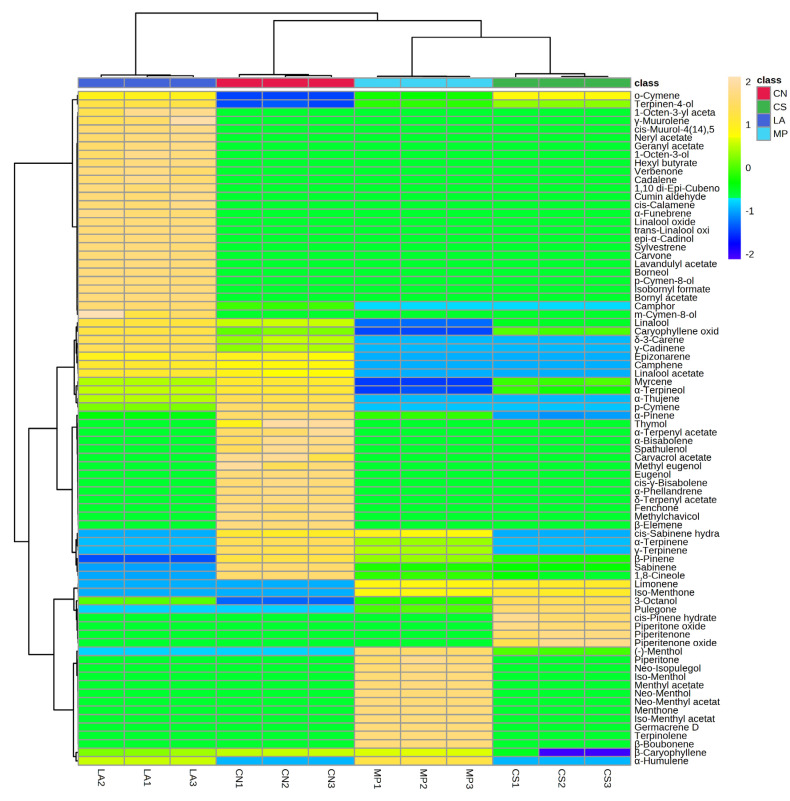
Heat map of the two-way hierarchical clustering analysis of detected phytochemicals. Abbreviations are as follows: CN: *Calamintha nepeta* Savi; CS: *Calamintha sylvatica* Bromf.; LA: *Lavandula austroapennina* N.G. Passal. Tundis & Upson; MP: *Mentha piperita* L.

**Figure 3 animals-14-00069-f003:**
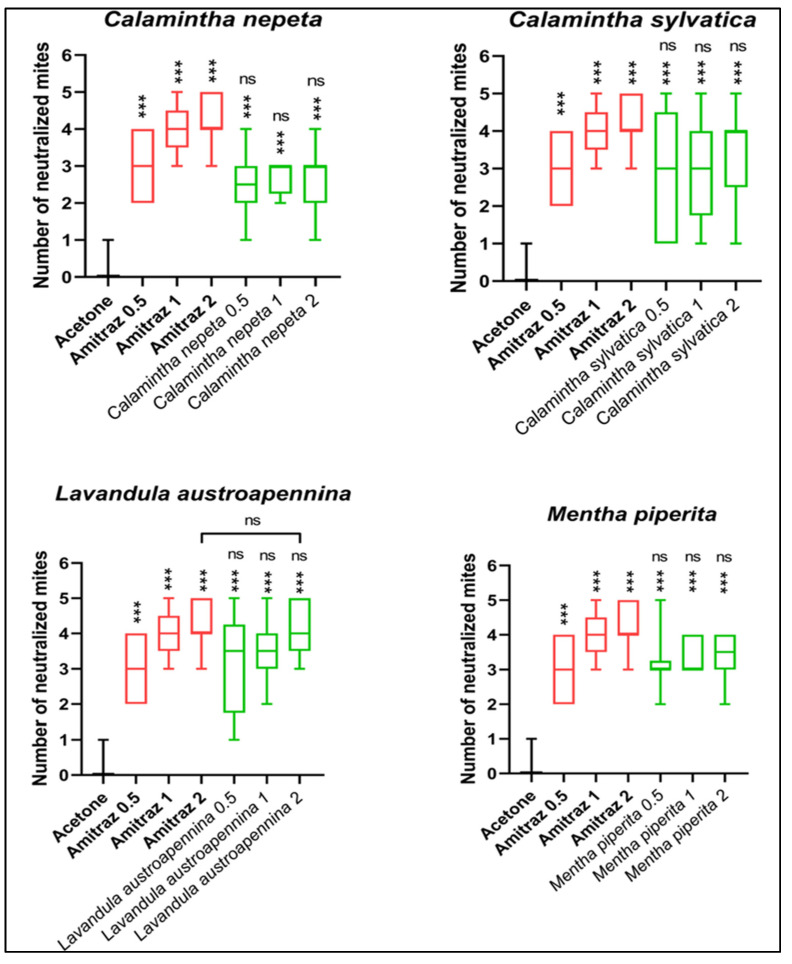
Percentages of the neutralization of the parasite *Varroa destructor* at doses of 0.5, 1 and 2 mg/mL with each EO, Acetone and Amitraz. *** *p* < 0.001 vs. acetone, ns (not significant) *p* > 0.05 vs. amitraz 0.5, amitraz 1.

**Figure 4 animals-14-00069-f004:**
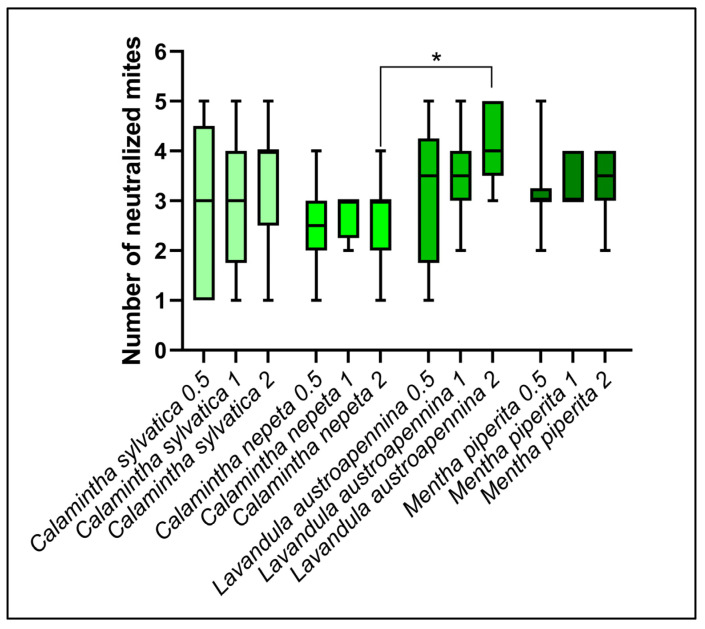
Comparison of the different EOs’ neutralization activity. * *p* < 0.05.

**Table 1 animals-14-00069-t001:** Chemical constituents of the investigated EOs.

N.	Compound ^1^	KI ^2^	KI ^3^	% ± SD				i.m. ^4^
				CN	CS	LA	MP	
1	α-Thujene	930	932	0.29 ± 0.00	-	0.11 ± 0.00	-	GC-MS
2	α-Pinene	939	937	4.82 ± 0.22	0.22 ± 0.00	0.47 ± 0.01	0.65 ± 0.00	GC, GC-MS
3	Camphene	954	955	0.18 ± 0.00	-	0.22 ± 0.00	-	GC, GC-MS
4	Sabinene	975	970	6.97 ± 0.07	0.22 ± 0.01	0.06 ± 0.00	0.26 ± 0.01	GC, GC-MS
5	β-Pinene	979	980	3.49 ± 0.13	0.44 ± 0.02	0.08 ± 0.00	0.92 ± 0.01	GC, GC-MS
6	1-Octen-3-ol	980	981	-	-	0.11 ± 0.00	-	GC-MS
7	Myrcene	991	990	0.63 ± 0.03	0.17 ± 0.01	0.30 ± 0.00	-	GC, GC-MS
8	3-Octanol	993	995	-	0.79 ± 0.02	0.11 ± 0.00	0.08 ± 0.00	GC-MS
9	α-Phellandrene	1002	1001	0.86 ± 0.03	-	-	-	GC, GC-MS
10	δ-3-Carene	1011	1010	0.13 ± 0.02	-	0.42 ± 0.00	-	GC-MS
11	α-Terpinene	1018	1015	0.27 ± 0.01	-	-	0.07 ± 0.00	GC, GC-MS
12	p-Cymene	1020	1020	0.53 ± 0.04	-	0.08 ± 0.00	-	GC, GC-MS
13	o-Cymene	1022	1021	-	0.49 ± 0.01	0.59 ± 0.01	0.11 ± 0.00	GC, GC-MS
14	Limonene	1029	1028	-	3.58 ± 0.02	-	3.02 ± 0.04	GC, GC-MS
15	Sylvestrene	1030	1030	-	-	1.42 ± 0.02	-	GC-MS
16	1,8-Cineole	1031	1031	34.09 ± 0.36	2.29 ± 0.39	1.11 ± 0.01	3.52 ± 0.08	GC, GC-MS
17	γ-Terpinene	1062	1061	0.36 ± 0.03	-	-	0.11 ± 0.00	GC, GC-MS
18	cis-Sabinene hydrate	1068	1070	0.12 ± 0.00	-	-	0.10 ± 0.00	GC, GC-MS
19	Linalool oxide	1072	1077	-	-	0.51 ± 0.00	-	GC-MS
20	trans-Linalool oxide	1086	1093	-	-	0.45 ± 0.01	-	GC-MS
21	Fenchone	1087	1090	2.15 ± 0.16	-	-	-	GC-MS
22	Terpinolene	1088	1089	-	-	-	0.08 ± 0.00	GC-MS
23	Linalool	1098	1101	6.64 ± 0.13	0.63 ± 0.03	26.67 ± 0.30	0.12 ± 0.00	GC, GC-MS
24	1-Octen-3-yl acetate	1112	1119	-	-	0.03 ± 0.01	-	GC-MS
25	cis-Pinene hydrate	1121	1120	-	0.09 ± 0.01	-	-	GC-MS
26	Camphor	1146	1142	0.08 ± 0.00	-	5.17 ± 0.04	-	GC, GC-MS
27	Neo-Isopulegol	1148	1149	-	-	-	0.28 ± 0.02	GC-MS
28	Menthone	1152	1158	-	-	-	21.67 ± 0.18	GC, GC-MS
29	Iso-Menthone	1162	1165	-	7.15 ± 0.27	-	5.88 ± 0.06	GC-MS
30	Neo-Menthol	1165	1167	-	-	-	4.07 ± 0.06	GS-MS
31	Borneol	1169	1164	-	-	15.24 ± 0.17	-	GC-MS
32	(-)-Menthol	1173	1174	-	1.33 ± 0.07	-	39.91 ± 0.41	GC, GC-MS
33	Terpinen-4-ol	1177	1175	0.25 ± 0.02	3.65 ± 0.14	17.44 ± 0.20	1.80 ± 0.02	GC, GC-MS
34	m-Cymen-8-ol	1180	1185	-	-	0.13 ± 0.05	-	GC-MS
35	Iso-Menthol	1182	1187	-	-	-	2.41 ± 0.08	GC-MS
36	p-Cymen-8-ol	1183	1188	-	-	1.44 ± 0.02	-	GC-MS
37	α-Terpineol	1188	1188	1.57 ± 0.00	0.59 ± 0.03	1.02 ± 0.00	0.24 ± 0.01	GC, GC-MS
38	Hexyl butyrate	1191	1202	-	-	0.81 ± 0.06	-	GC-MS
39	Methylchavicol	1195	1195	1.10 ± 0.04	-	-	-	GC-MS
40	Verbenone	1204	1206	-	-	0.23 ± 0.02	-	GC-MS
41	Isobornyl formate	1233	1234	-	-	1.07 ± 0.01	-	GC, GC-MS
42	Pulegone	1237	1235	-	20.91 ± 0.54	-	0.78 ± 0.06	GC, GC-MS
43	Cumin aldehyde	1239	1236	-	-	0.34 ± 0.01	-	GC-MS
44	Carvone	1242	1239	-	-	0.46 ± 0.00	-	GC-MS
45	Piperitone	1249	1252	-	-	-	1.54 ± 0-17	GC-MS
46	Linalool acetate	1257	1256	11.25 ± 1.07	-	14.67 ± 0.30	-	GC, GC-MS
47	Neo-Menthyl acetate	1273	1281	-	-	-	0.54 ± 0.00	GC-MS
48	Bornyl acetate	1288	1289	-	-	0.09 ± 0.00	-	GC, GC-MS
49	Piperitone oxide	1288	1290	-	37.70 ± 2.08	-	-	GC-MS
50	Lavandulyl acetate	1289	1302	-	-	0.28 ± 0.00	-	GC-MS
51	Thymol	1290	1271	0.78 ± 0.40	-	-	-	GC, GC-MS
52	Menthyl acetate	1295	1303	-	-	-	7.56 ± 0.16	GC, GC-MS
53	Iso-Menthyl acetate	1305	1313	-	-	-	0.12 ± 0.00	GC-MS
54	δ-Terpenyl acetate	1317	1322	0.16 ± 0.00	-		-	GC-MS
55	Piperitenone	1343	1342	-	0.55 ± 0.05	-	-	GC-MS
56	α-Terpenyl acetate	1350	1355	0.75 ± 0.12	-	-	-	GC-MS
57	Eugenol	1356	1359	14.66 ± 0.19	-	-	-	GC, GC-MS
58	Piperitenone oxide	1363	1365	-	18.26 ± 4.03	-	-	GC-MS
59	Neryl acetate	1365	1369	-	-	0.15 ± 0.00	-	GC, GC-MS
60	Carvacrol acetate	1372	1376	0.46 ± 0.12	-	-	-	GC, GC-MS
61	Geranyl acetate	1381	1385	-	-	0.19 ± 0.01	-	GC, GC-MS
62	β-Boubonene	1388	1388	-	-	-	0.17 ± 0.00	GC-MS
63	β-Elemene	1390	1401	0.40 ± 0.00	-	-	-	GC-MS
64	α-Funebrene	1397	1397	-	-	0.11 ± 0.00	-	GC-MS
65	Methyl eugenol	1403	1406	1.92 ± 0.40	-	-	-	GC, GC-MS
66	β-Caryophyllene	1408	1406	2.89 ± 0.06	0.63 ± 0.01	2.35 ± 0.17	3.21 ± 0.03	GC, GC-MS
67	α-Humulene	1454	1445	-	-	0.06 ± 0.00	0.13 ± 0.00	GC-MS
68	cis-Muurol-4(14),5-diene	1460	1457	-	-	0.49 ± 0.04	-	GC-MS
69	γ-Muurolene	1479	1476	-	-	0.09 ± 0.03	-	GC-MS
70	Germacrene D	1480	1475	-	-	-	0.63 ± 0.00	GC-MS
71	Epizonarene	1497	1500	0.15 ± 0.01	-	0.17 ± 0.02	-	GC-MS
72	α-Bisabolene	1505	1511	0.40 ± 0.06	-	-	-	GC, GC-MS
73	γ-Cadinene	1513	1510	0.30 ± 0.04	-	1.13 ± 0.02	-	GC-MS
74	cis-γ-Bisabolene	1515	1500	0.28 ± 0.01	-	-	-	GC-MS
75	cis-Calamene	1521	1519	-	-	0.65 ± 0.03	-	GC-MS
76	Spathulenol	1576	1578	0.35 ± 0.05	-	-	-	GC-MS
77	Caryophyllene oxide	1583	1569	0.39 ± 0.04	0.32 ± 0.01	1.34 ± 0.03	-	GC-MS
78	1,10 di-Epi-Cubenol	1614	1600	-	-	0.40 ± 0.01	-	GC-MS
79	epi-α-Cadinol	1640	1631	-	-	0.90 ± 0.01	-	GC-MS
80	Cadalene	1674	1645	-	-	0.07 ± 0.00	-	GC-MS

^1^ Components are reported according to their elution order on an apolar column; ^2^ KI from the literature; ^3^ KI measured on an apolar column; ^4^ Identification method. CN *Calamintha nepeta* Savi; CS: *Calamintha sylvatica* Bromf.; LA: *Lavandula austroapennina* N.G. Passal. Tundis & Upson; MP: *Mentha piperita* L.

**Table 2 animals-14-00069-t002:** Efficacy percentages (%) with the standard deviation (±) of the essential oils.

Concentration mg/mL	*C. nepeta* % Death	*C. sylvatica* % Death	*L. austroapennina* % Death	*M. piperita* % Death	Acetone % Death	Amitraz % Death
0.5 mg	38 (±29)	48 (±37)	64 (±30)	64 (±16)	2 (±5)	60 (±20)
1 mg	54 (±25)	58 (±27)	70 (±17)	68 (±10)		66 (±35)
2 mg	52 (±17)	60 (±31)	80 (±21)	68 (±14)		94 (±10)

## Data Availability

Data are kept at Magna Græcia University of Catanzaro and are available upon request.

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
