# Peer review of "Phytochemical Composition and Pharmacological Efficacy Evaluation of Calamintha nepeta, Calamintha sylvatica, Lavandula austroapennina and Mentha piperita Essential Oils for the Control of Honeybee (Apis mellifera) Varroosis"

_animals, 2023, doi:10.3390/ani14010069_

Round 1
Reviewer 1 Report
Comments and Suggestions for Authors
Comments to authors
Overall: There is a considerable number of grammatical errors in the text.
Introduction:
Overall, the introduction focuses a lot on the negative effects of Varroa on honeybees. However, the main premise of this article is to investigate new control methods. Thus, I would wish to read more information about the current control methods that are being used. There are other naturally derived chemical control options (e.g., oxalic acid or formic acid) – do we need other options? There wasn’t a strong point to support your study.
Line 51: I do not think it is necessary to add «(V. destructor)»
Line 52: Varroa feeds on the “immature stages” of honey bees, and not just the larval stage
Line 52 – 56: I would suggest rewriting this section to something along the line of:
“Adult female V. destructor reproduce in capped brood cells, and feed on the fat body of honey bee pupae. Along with her offspring, she will emerge with the developed bee and subsequently, they will continue the biological cycle. In the search of developing bees, V. destructor attach to adult nurse bees and take advantage of them visiting brood.”
Line 57-58: I would remove the sentence “It is estimated that […]”
Line 59: How can the body weight be “reduced from 6.3% to 25%”, if the second number is greater?
Line 63: I would suggest removing “reducing their ability to defend themselves and” – I think it is redundant.
Line 64 – 65: I think it is irrelevant to mention the FAD- glucosodehydrogenase and suggest removing these two sentences.
Line 65: In the case of Varroa, a pupa is not “infected” but infested. Please adjust.
Line 66: “Born parasite honeybees” is not correct. I think what you mean is “When emerging, parasitized honeybees have half as much […]”
Line 68-69: The syndrome is called “parasitic mite syndrome”. Also, there are no honey bee colonies where Varroa is not present constantly. The difference lays in the number of mites, or the mite pressure. Please adjust the statement accordingly.
Line 69: This sentence is grammatically incorrect. It would be: “This syndrome is associated with the transmission of various viruses by mites [9,10].”
Line 70: The sentence “The activity of the viral vector […]” does not make sense – it is not the vector that manifests these symptoms but rather the virus itself.
Line 71: How is “malformation of wings” different from “deformed wings”?
Line 75: I disagree that chemical control is a “quick fix” for the issues that Varroa causes. Why are we then still struggling with this parasite?
Line 79-80: Is it true that resistance cannot be prevented at all?
Line 82: Adjust the wording; coumaphos does not itself “move” into the hive. It is distributed throughout the hive.
Line 103 – 105: Why did you choose these plant species? This is not clear.
Material and Methods:
Overall, I find it problematic that you exposed both mites and bees to your test acaricides in a very artificial manner. In a hive setting, the EO’s would act as volatiles, but it seems with your experimental design they rather act via contact; unless I misunderstood. Also, I am not sure whether exposing mites or bees for 1 hour counts as “acute toxicity testing”.
Line 151-152: I think “positive and negative control” are in reversed order in this sentence. Amitraz would be the positive control.
Line 168: Did you test whether the EO’s are still present in the tube, and at what concentration?
Line 170: Why 5 mites?
Line 174-175: Exposing mites for 1 hour doesn’t seem to be enough to measure acute toxicity. Standard acute toxicity assays last for 24 hours at least.
Line 178 – 179: Even if “inactive” after 1 hour, isn’t it possible that these mites would recover later?
Line 189: Why did you only place 5 bees in a falcon tube if you collected 25 from the start?
Line 197: What was the level used to identify statistical significance?
Line 204: What data was the multivariate analysis used for? What data was the PCA used for? This is not clear.
Line 209: What does it mean that you “pretreated” the data?
Results:
Overall: I suggest that table 1 should go into supplementary materials. Also, I think you should compare the efficacy re. mite toxicity AMONG the different EO’s you tested, instead of only looking at them individually. Ideally, you can statistically support your claim of in Lie 355-256 that one EO is better than others.
Line 216 – 217: This sentence should not be in the results.
Line 266: I am not sure how valuable this heat map is.
Line 280: This is not a table, but a figure. Also, I do not understand why you have added “ns” to some boxplots but not others.
Line 285-293: By looking at the figure, these statements do not make any sense. It seems that all EO’s are performing better than Acetone along, but not any different than any of the Amitraz concentrations.
Line 295: You need to support this statement with showing the data.
Discussion:
Line 297 – 305: I do not see how genetics are relevant for your discussion. I would remove this altogether.
Line 312: I don’t see how human health is relevant here.
Line 330-335: So, it seems that the tested EO’s do differ extensively in their composition. How do you explain that they still all seem to results in a similar toxicity to mites? Which metabolites are most likely responsible for that?
Line 360: Can you refer to “killing rate”? I think you would have to talk about “neutralizing rate”.
Comments on the Quality of English Language
There are a lot of grammatical errors, missing articles and errors in sentence structure.
Author Response
Dear reviewer, thank you very much for the attention paid to our manuscript and for the information given. Below you will find the answers, point by point, to your questions. They are written in bold.
Overall, the introduction focuses a lot on the negative effects of Varroa on honeybees. However, the main premise of this article is to investigate new control methods. Thus, I would wish to read more information about the current control methods that are being used. There are other naturally derived chemical control options (e.g., oxalic acid or formic acid) – do we need other options? There wasn’t a strong point to support your study.
R: Thank you for your comments which helps to improve the overall quality of the manuscript. First, the English was revised by a native speaker. About this first comment, we have rewritten the introductory part by adding paragraphs dealing with other environmentally friendly control methods, such as organic acids. We pointed out that although these control methods have many positive effects, they are not without side effects. These assertions bring more strength to our study.
1) Line 51: I do not think it is necessary to add «(V. destructor)»
R: It was deleted accordingly
2) Line 52: Varroa feeds on the “immature stages” of honey bees, and not just the larval stage
R: Thanks for your comment, the phrase was corrected accordingly
3) Line 52 – 56: I would suggest rewriting this section to something along the line of:
“Adult female V. destructor reproduce in capped brood cells, and feed on the fat body of honey bee pupae. Along with her offspring, she will emerge with the developed bee and subsequently, they will continue the biological cycle. In the search of developing bees, V. destructor attach to adult nurse bees and take advantage of them visiting brood.”
R: Thank you for your comment, the sentence has been reworked along the lines of your suggestion
4) Line 57-58: I would remove the sentence “It is estimated that […]”
R: Now amended
5) Line 59: How can the body weight be “reduced from 6.3% to 25%”, if the second number is greater?
R: Thanks for your comment, the sentence has been rewritten to make the concept clearer.
6) Line 63: I would suggest removing “reducing their ability to defend themselves and” – I think it is redundant.
R: Removed accordingly to your advice
7) Line 64 – 65: I think it is irrelevant to mention the FAD- glucosodehydrogenase and suggest removing these two sentences.
R: Removed accordingly to your device
8) Line 65: In the case of Varroa, a pupa is not “infected” but infested. Please adjust.
R: Now amended
9) Line 66: “Born parasite honeybees” is not correct. I think what you mean is “When emerging, parasitized honeybees have half as much […]”
R: Thanks for your comment; the sentence has been rewritten to make the concept clearer
10) Line 68-69: The syndrome is called “parasitic mite syndrome”. Also, there are no honey bee colonies where Varroa is not present constantly. The difference lays in the number of mites, or the mite pressure. Please adjust the statement accordingly.
R: Thanks for your comment; the sentence has been rewritten to make the concept clearer
11) Line 69: This sentence is grammatically incorrect. It would be: “This syndrome is associated with the transmission of various viruses by mites [9,10].”
R: The sentence has been corrected accordingly
12) Line 70: The sentence “The activity of the viral vector […]” does not make sense – it is not the vector that manifests these symptoms but rather the virus itself.
R: Thanks for your comment. The sentence has been rewritten to make the concept clearer
13) Line 71: How is “malformation of wings” different from “deformed wings”?
R: Thank you for your comment, we have deleted this repetition.
14) Line 75: I disagree that chemical control is a “quick fix” for the issues that Varroa causes. Why are we then still struggling with this parasite?
R: The sentence has been reworded to make the concept clearer, what we meant to say is that the synthetic treatment is often easier and quicker to apply
15) Line 79-80: Is it true that resistance cannot be prevented at all?
R: The concept has been expressed differently in the revised version.
16) Line 82: Adjust the wording; coumaphos does not itself “move” into the hive. It is distributed throughout the hive.
R: Thanks for your comment. The sentence has been reworded to make the concept clearer
17) Line 103 – 105: Why did you choose these plant species? This is not clear.
R: Thank you for your comment. We have provided more detail on our choice
Material and Methods:
18) Overall, I find it problematic that you exposed both mites and bees to your test acaricides in a very artificial manner. In a hive setting, the EO’s would act as volatiles, but it seems with your experimental design they rather act via contact; unless I misunderstood. Also, I am not sure whether exposing mites or bees for 1 hour counts as “acute toxicity testing”.
R: The one used in this study is a residual contact toxicity study that serves as a preliminary screening to validate the efficacy of new molecules/compounds against different pest species. In several previous published studies on Varroa (please, see for e.g. https://doi.org/10.1093/jee/99.5.1579, https://doi.org/10.3896/IBRA.1.48.4.06, https://doi.org/10.3390/insects12111045, https://doi.org/10.3390/pathogens10091182, etc.), this method has been used. Exposing mites to times longer than one hour could lead to a bias in the results and their interpretation. Indeed, Milani et al., 2001, (https://doi.org/10.1051/apido:2001118) noted that mites suffer from artificial conditions and if they are removed from their environment for more than 4 hours, they may die; such an eventuality would not allow us to understand whether the deaths are due to the active ingredient tested or to the artificial conditions.
19) Line 151-152: I think “positive and negative control” are in reversed order in this sentence. Amitraz would be the positive control.
R: Thank you very much for your comment pointing out this typo. It has now been corrected.
20) Line 168: Did you test whether the EO’s are still present in the tube, and at what concentration?
R: Thank you for your comment. In response, I quote a sentence from Gashout et al. (2009), doi: 10.3896/IBRA.1.48.4.06: “It is unlikely that a significant amount of the products tested would have evaporated within this time period, due to their high boiling points, all above 200ºC”. You can also find further information in CRC (2007) CRC handbook of chemistry and physics (88th ed.). CRC Press; Boca Raton, FL, USA. pp. 2640.
21) Line 170: Why 5 mites?
R: It is common to do this test with five mites. Please, see Gashout et al. (2009), doi: 10.3896/IBRA.1.48.4.06; Hybl et al. (2021) Insects 2021, 12, 1045; Bava et al. (2021) Pathogens. 2021; 10(9):1182
22) Line 174-175: Exposing mites for 1 hour doesn’t seem to be enough to measure acute toxicity. Standard acute toxicity assays last for 24 hours at least.
R: Please refer to the articles in which residual toxicity tests have been carried out against Varroa destructor. The residual toxicity test is extended for a maximum of 4 hours. Milani et al, 2001, (https://doi.org/10.1051/apido:2001118) highlighted that varroa suffers artificial conditions, even 4 hours of removal from its environment. The 4 hour time was already brought to 2 hours by Hybl et al. 2021 (https://doi.org/10.3390/insects12111045) and 1 hour by us in our previous study, Bava et al. (2021) Pathogens. 2021; 10(9):1182.
23) Line 178 – 179: Even if “inactive” after 1 hour, isn’t it possible that these mites would recover later?
R: Thanks for your comment. We checked whether the mites recovered or not. The mites do not recover after exposure. We had already done the same in our previous study, namely Bava et al. (2021) Pathogens. 2021; 10(9):1182.
24) Line 189: Why did you only place 5 bees in a falcon tube if you collected 25 from the start?
R: Thanks for your comment. The period has been rewritten to make the procedure better understandable.
25) Line 197: What was the level used to identify statistical significance?
R: In the "statistical analysis" paragraph you can find it written that the result is considered statistically significant with a p value less than 0.05.
26) Line 204: What data was the multivariate analysis used for? What data was the PCA used for? This is not clear.
R: thanks for this suggestion. We tried to better describe this section in the revised text.
Multivariate analysis was used to highlight the different distribution of the phytochemical constituents in the investigated essential oils. The multivariate approach of principal component analysis (PCA) was chosen in this study. PCA is a multivariate statistical method widely used to identify the relationship between the original indicator variables and transform them into independent principal components (Reference: Yang, W., Zhao, Y., Wang, D., Wu, H., Lin, A., & He, L. (2020). Using principal components analysis and IDW interpolation to determine spatial and temporal changes of surface water quality of Xin’anjiang river in Huangshan, China. International journal of environmental research and public health, 17(8), 2942).
The data matrix consisted of 12 samples (three analyses for each EO) and 80 variables (the percentage abundance of identified secondary metabolites).
27) Line 209: What does it mean that you “pretreated” the data?
R: thanks for this question. As described by Van den Berg and coworkers, selecting a proper data pretreatment method is an essential step in the analysis of metabolomic data. This is due to different factors which may occur, such as the differences in orders of magnitude between measured metabolite concentrations, or technical variation originating from sampling and analytical errors. As reported, different data pretreatment methods, such as centering, autoscaling, pareto scaling, range scaling, or log transformation may be used.
(Reference: Van den Berg, R. A., Hoefsloot, H. C., Westerhuis, J. A., Smilde, A. K., & Van der Werf, M. J. (2006). Centering, scaling, and transformations: improving the biological information content of metabolomics data. BMC genomics, 7, 1-15.).
Results:
28) Overall: I suggest that table 1 should go into supplementary materials. Also, I think you should compare the efficacy re. mite toxicity AMONG the different EO’s you tested, instead of only looking at them individually. Ideally, you can statistically support your claim of in Lie 355-256 that one EO is better than others.
R: Thank you very much for this important suggestion that helps us improve the overall quality of the manuscript. The statistics have been expanded according to your important advice and the acaricidal efficacy of essential oils was compared with each other. With reference to table 1, with your consent, we would like to keep it in the text as we consider it of particular importance for the characterization of the tested oils.
29) Line 216 – 217: This sentence should not be in the results.
R: The sentence was deleted.
30) Line 266: I am not sure how valuable this heat map is.
R: We previously added this heat map to offer a clear graphical overview of the distribution of EO’s chemical constituents. According to your suggestion, this figure has been deleted in the revised paper.
31) Line 280: This is not a table, but a figure. Also, I do not understand why you have added “ns” to some boxplots but not others.
R: Thank you for pointing out this oversight concerning the figure. We have corrected it. Regarding the ns specification, we have corrected the figure adding more specification about the statistical significance. Hope that is sufficient to better understand the results.
32) Line 285-293: By looking at the figure, these statements do not make any sense. It seems that all EO’s are performing better than Acetone along, but not any different than any of the Amitraz concentrations.
R: The statement is correct. All essential oils are more effective than acetone (negative control), they are statistically no different from amitraz (positive control). This last finding is of great value not to be considered negative. In fact, amitraz is the synthetic positive control, those we tested are compounds of natural origin.
33) Line 295: You need to support this statement with showing the data.
R: No data were presented because none of the bees subjected to treatment died.
Discussion:
34) Line 297 – 305: I do not see how genetics are relevant for your discussion. I would remove this altogether.
R: The sentence has been deleted as suggested
35) Line 312: I don’t see how human health is relevant here.
R: The sentence has been deleted as suggested
36) Line 330-335: So, it seems that the tested EO’s do differ extensively in their composition. How do you explain that they still all seem to results in a similar toxicity to mites? Which metabolites are most likely responsible for that?
R: Line 330-335: Their effectiveness, as you underlined, depends on their composition and this would explain the different toxicity on the V. destructor mite. Toxicity is presumably attributable to terpenes, monoterpenes and terpenoids, which are present in varying concentrations in different oils. To clarify better, we have added the table below (Table 2) in the text which shows the efficacy percentages.
Table 2. Efficacy percentages (%) with standard deviation (±) of the essential oils
Concentration mg/mL |
C. nepeta % death |
C. sylvatica % death |
L. austroapennina % death |
M. piperita % death |
Acetone % death |
Amitraz % death |
0.5 mg |
38 (±29) |
48 (±37) |
64 (±30) |
64 (±16) |
2 (±5) |
60 (±20) |
1 mg |
54 (±25) |
58 (±27) |
70 (±17) |
68 (±10) |
66 (±35) |
|
2 mg |
52 (±17) |
60 (±31) |
80 (±21) |
68 (±14) |
94 (±10) |
37) Line 360: Can you refer to “killing rate”? I think you would have to talk about “neutralizing rate”.
R: Now amended as suggested

Reviewer 2 Report
Comments and Suggestions for Authors
This manuscript (animals-2733864) explored on the possible acaricide action of four essential oils extracted from botanical species (Lavandula austroapennina, Calamintha nepeta, Calamintha sylvatica and Mentha piperita) native to the Calabria region by means of contact residual toxicity tests. The results showed that the lavender proved to have the best acaricidal activity, achieving an 80% mite-killing rate at 2 mg/mL. This study provides valuable information for the control of Varroa destructor parasitosis. The manuscript is written generally. And the data is sufficient to support its main conclusion. It meets in the present form the criteria to be published in the journal. However, major revision should be addressed prior to publication.
The manuscript needs careful editing by someone with expertise in technical English so that the goals of the results of this study are clear to a reader.
What is the relationship between the chemical constituents of investigated Eos and acaricidal activity? Why not determine the acaricidal activity of each isolate (table 1) against Varroa destructor?
Line 195, provide data on the toxicity of essential oil (Lavandula austroapennina, Calamintha nepeta, Calamintha sylvatica and Mentha piperita) to bees, as well as data on the growth and development of bees after treatment with different concentrations, since 2 mg/mL is still a very high concentration,
Comments on the Quality of English Language
Formatting problems in manuscript should be modified carefully.
Author Response
Dear reviewer, thank you very much for your attention to our manuscript and for the information you gave. We have answered your questions point by point. You will find the answers below written in bold.
This manuscript (animals-2733864) explored on the possible acaricide action of four essential oils extracted from botanical species (Lavandula austroapennina, Calamintha nepeta, Calamintha sylvatica and Mentha piperita) native to the Calabria region by means of contact residual toxicity tests. The results showed that the lavender proved to have the best acaricidal activity, achieving an 80% mite-killing rate at 2 mg/mL. This study provides valuable information for the control of Varroa destructor parasitosis. The manuscript is written generally. And the data is sufficient to support its main conclusion. It meets in the present form the criteria to be published in the journal. However, major revision should be addressed prior to publication.
Response: Thanks for your valuable work addressing to the improvement of our manuscript. The English was revised by a native speaker. The revisions are highlighted in the text
1)The manuscript needs careful editing by someone with expertise in technical English so that the goals of the results of this study are clear to a reader.
R: Many thanks for this comment. The manuscript was revised by a native speaker
2) What is the relationship between the chemical constituents of investigated Eos and acaricidal activity? Why not determine the acaricidal activity of each isolate (table 1) against Varroa destructor?
R: Thank you for your observation which helps us explain a concept to which we are particularly attached. We have tested the essential oil according to methods published by various research groups, please see: Gashout et al. (2009), doi: 10.3896/IBRA.1.48.4.06; Hybl et al. (2021) Insects 2021, 12, 1045; Bava et al. (2021) Pathogens. 2021; 10(9):1182. The essential oil is a phytocomplex and as such all the components act in synergy to determine the acaricidal activity. For this reason, the entire EO and not isolated constituents were tested.
3) Line 195, provide data on the toxicity of essential oil (Lavandula austroapennina, Calamintha nepeta, Calamintha sylvatica and Mentha piperita) to bees, as well as data on the growth and development of bees after treatment with different concentrations, since 2 mg/mL is still a very high concentration.
R: We followed the guidelines of the article published by Hybl et al., 2021, Insects 2021, 12, 1045, in which only acaricidal effiacia was tested. The aim of this work was to assess the toxicity of essential oils towards Varroa; the test on adult honeybees was an additional test we wanted to support our study, which was already used in the following study: Bava et al. (2021) Pathogens. 2021; 10(9):1182. This test produced no mortality in the honey bee samples tested.
Dear reviewer, thank you very much for your attention to our manuscript and for the information you gave. We have answered your questions point by point. You will find the answers below written in red.
This manuscript (animals-2733864) explored on the possible acaricide action of four essential oils extracted from botanical species (Lavandula austroapennina, Calamintha nepeta, Calamintha sylvatica and Mentha piperita) native to the Calabria region by means of contact residual toxicity tests. The results showed that the lavender proved to have the best acaricidal activity, achieving an 80% mite-killing rate at 2 mg/mL. This study provides valuable information for the control of Varroa destructor parasitosis. The manuscript is written generally. And the data is sufficient to support its main conclusion. It meets in the present form the criteria to be published in the journal. However, major revision should be addressed prior to publication.
Response: Thanks for your valuable work addressing to the improvement of our manuscript. The English was revised by a native speaker. The revisions are highlighted in the text
1)The manuscript needs careful editing by someone with expertise in technical English so that the goals of the results of this study are clear to a reader.
R: Many thanks for this comment. The manuscript was revised by a native speaker
2) What is the relationship between the chemical constituents of investigated Eos and acaricidal activity? Why not determine the acaricidal activity of each isolate (table 1) against Varroa destructor?
R: Thank you for your observation which helps us explain a concept to which we are particularly attached. We have tested the essential oil according to methods published by various research groups, please see: Gashout et al. (2009), doi: 10.3896/IBRA.1.48.4.06; Hybl et al. (2021) Insects 2021, 12, 1045; Bava et al. (2021) Pathogens. 2021; 10(9):1182. The essential oil is a phytocomplex and as such all the components act in synergy to determine the acaricidal activity. For this reason, the entire EO and not isolated constituents were tested.
3) Line 195, provide data on the toxicity of essential oil (Lavandula austroapennina, Calamintha nepeta, Calamintha sylvatica and Mentha piperita) to bees, as well as data on the growth and development of bees after treatment with different concentrations, since 2 mg/mL is still a very high concentration.
R: We followed the guidelines of the article published by Hybl et al., 2021, Insects 2021, 12, 1045, in which only acaricidal effiacia was tested. The aim of this work was to assess the toxicity of essential oils towards Varroa; the test on adult honeybees was an additional test we wanted to support our study, which was already used in the following study: Bava et al. (2021) Pathogens. 2021; 10(9):1182. This test produced no mortality in the honey bee samples tested.

Reviewer 3 Report
Comments and Suggestions for Authors
The topic of this research paper seems appealing, with research showing that among the essential oils tested, lavender may be a good candidate for follow-up evaluation. Unfortunately, it appears to contain many inaccuracies that call into question the authenticity of the entire document. Here are some examples and questions.
1. "This means that on average the weight of nascent Varroa-infested honeybees is reduced from 25% to 6.3%." (See lines 58) The values should be reversed.
2. "Acetone and Amitraz served as negative and positive controls, respectively." (See lines 151-152) The order of the controls should be reversed.
3. "Molecules 2020, 25." The font for "2020" should be thickened. (See lines 423)
4. "Routledge, 2020; pp. 113–157." The font for "2020" should be thickened. (See lines 431)
Author Response
Dear reviewer, thank you very much for your attention to our manuscript and for the information you gave. We have answered your questions point by point. You will find the answers below written in bold.
The topic of this research paper seems appealing, with research showing that among the essential oils tested, lavender may be a good candidate for follow-up evaluation. Unfortunately, it appears to contain many inaccuracies that call into question the authenticity of the entire document. Here are some examples and questions.
1) "This means that on average the weight of nascent Varroa-infested honeybees is reduced from 25% to 6.3%." (See lines 58) The values should be reversed.
R: Thank you for your comment which brings to light this concept which is not explained as we would have liked. The numbers should not be reversed; the sentence has been reformulated to better explain the concept.
2) "Acetone and Amitraz served as negative and positive controls, respectively." (See lines 151-152) The order of the controls should be reversed.
R: Thanks for your comment. The text has been corrected accordingly
3) "Molecules 2020, 25." The font for "2020" should be thickened. (See lines 423)
R: Now amended
4) "Routledge, 2020; pp. 113–157." The font for "2020" should be thickened. (See lines 431)
R: Now amended

Reviewer 4 Report
Comments and Suggestions for Authors
General comment: Also essential oils are "chemical compounds", it's better to use the term "synthetic compounds" or similar. It's only for them also that resistance has been reported. For the compounds from natural origin (thymol, but also formic and oxalic acid) no resistance is reported. The substances of natural origin are widely used, so the categoric statement is incorrect. So, to avoid inaccuracies, it's better to use a more precise term.
Line 66: Better: Honeybees born after parasitation in the larval stage...
Line 67: not "normal"... better "unparasitized bees".
For the introduction: Better define your aim with this study. "Native to the Calabria region" is valid, but independently if native or not, it's definitely necessary to go through the registration process later on etc. So, a better argument is to give more alternatives etc. You're at the first step of the development of a treatment, but it would be better to make clear that you know the whole process from the beginning.
Lines 305-306: Again, don't overplay the role of resistance. More alternatives are necessary and important, but you sound as they were any. This is not correct.
Lines 313 ff: "Recently" isn't really correct if you cite a paper from 1997. Before that, thymol treatments were developed at Liebefeld (the group around Imdorf, Bogdanov etc.), so essential oils have been used against varroa since the 1990s. There are several products registered with this compound. I assume that this is inaccurate wording, not lack of knowledge of the existing products/market. This work is also cited/described in the paper of Wallner 1999 or in Rosenkranz et al. 2010.
This is promising and interesting research that definitely deserves be carried on. In the discussion, however, it should be made clear that the authors are aware that they are talking about veterinary medicine. There's a guideline that regulates how to develop varroacides (Guideline on veterinary medicinal products controlling Varroa destructor parasitosis in bees (europa.eu)) which should be at least cited. Also it should be discussed that the measured efficicacy isn't sufficient (stabilized with at least 90% for substances of natural origin in this guideline). This can be tackled by the formulation, but I miss noticing that the authors are aware of this.
Comments on the Quality of English Language
English is generally good - only pay attention that it doesn't get too colloquial ("normal bees").
Author Response
Dear reviewer, thank you very much for your attention to our manuscript and for the information you gave. We have answered your questions point by point. You will find the answers below written in bold.
General comment: Also essential oils are "chemical compounds", it's better to use the term "synthetic compounds" or similar. It's only for them also that resistance has been reported. For the compounds from natural origin (thymol, but also formic and oxalic acid) no resistance is reported. The substances of natural origin are widely used, so the categoric statement is incorrect. So, to avoid inaccuracies, it's better to use a more precise term.
Response:
1)Line 66: Better: Honeybees born after parasitation in the larval stage...
R: Thanks for your advice. The text has been corrected accordingly
2) Line 67: not "normal"... better "unparasitized bees".
R: Thanks for your advice. The text has been corrected accordingly
3) For the introduction: Better define your aim with this study. "Native to the Calabria region" is valid, but independently if native or not, it's definitely necessary to go through the registration process later on etc. So, a better argument is to give more alternatives etc. You're at the first step of the development of a treatment, but it would be better to make clear that you know the whole process from the beginning.
R: Thanks for your comment that help us to improve the overall quality of the manuscript. More reasons for our choice of these essential oils have been given in the introduction
4) Lines 305-306: Again, don't overplay the role of resistance. More alternatives are necessary and important, but you sound as they were any. This is not correct.
R: Thanks for your comment. More details have been provided regarding the treatment alternatives present, their strengths and weaknesses, and why we chose to investigate essential oils.
5) Lines 313 ff: "Recently" isn't really correct if you cite a paper from 1997. Before that, thymol treatments were developed at Liebefeld (the group around Imdorf, Bogdanov etc.), so essential oils have been used against varroa since the 1990s. There are several products registered with this compound. I assume that this is inaccurate wording, not lack of knowledge of the existing products/market. This work is also cited/described in the paper of Wallner 1999 or in Rosenkranz et al. 2010.
R: Thank you for highlighting this inaccuracy to us. The text has been modified according to your advice.
6) This is promising and interesting research that definitely deserves be carried on. In the discussion, however, it should be made clear that the authors are aware that they are talking about veterinary medicine. There's a guideline that regulates how to develop varroacides (Guideline on veterinary medicinal products controlling Varroa destructor parasitosis in bees (europa.eu)) which should be at least cited. Also it should be discussed that the measured efficicacy isn't sufficient (stabilized with at least 90% for substances of natural origin in this guideline). This can be tackled by the formulation, but I miss noticing that the authors are aware of this.
R: Thank you for this important comment which helps us improve discussions. The guidelines you recommended have been incorporated into the text. A sentence has been added on the efficacy that a natural product must attest to according to the guidelines, as well as stating that the achievement of superior acaricide characteristics can be achieved by working on the formulations.

Round 2
Reviewer 2 Report
Comments and Suggestions for Authors
The authors have made modifications according to the modification comments. They allocated substantial effort to improving the quality of the manuscript. Thus, this manuscript “Phytochemical composition and pharmacological efficacy evaluation of Calamintha nepeta, Calamintha sylvatica, Lavandula austroapennina and Mentha piperita essential oils for the control of honeybee (Apis mellifera) varroosis” could be officially accepted for publication.
Author Response
Dear reviewer,
thank you very much for appreciating our effort and thank you again for the suggestions given that allowed us to improve the manuscript.
Reviewer 4 Report
Comments and Suggestions for Authors
I don't have any further comments.
Author Response
Dear reviewer,
thank you very much for appreciating the new version of the manuscript, and thank you again for your guidance in improving the paper.